

**Pelagic coccolithophore production and dissolution and their impacts on particulate**
**inorganic carbon cycling in the western North Pacific**
Yuye Han[1,2], Zvi Steiner[2], Zhimian Cao[1*], Di Fan[3], Junhui Chen[1], Jimin Yu[3] and Minhan Dai[1*]
[1]State Key Laboratory of Marine Environmental Science & College of Ocean and Earth Sciences, Xiamen University, Xiamen,
China
[2]Marine Biogeochemistry Division, GEOMAR Helmholtz Centre for Ocean Research, Kiel, Germany
[3]Laoshan Laboratory, Qingdao, China
*Correspondence to*: mdai@xmu.edu.cn & zmcao@xmu.edu.cn



**Abstract.** Coccolithophores, a type of single-celled phytoplankton that is abundant in global oceans, are closely associated
with the carbonate pump and thus play a crucial role in the marine carbon cycle. Here we investigated coccolithophore
abundances, species compositions, coccolithophore calcium carbonate ($CaCO_3$ as calcite) and particulate inorganic carbon
(PIC) concentrations in the upper water column of the western North Pacific Ocean, along a meridional transect spanning the
oligotrophic subtropical gyre and the nutrient-richer Kuroshio-Oyashio transition region. Our results revealed that
*Umbellosphaera tenuis* was the dominant coccolithophore species in the former, while *Emiliania huxleyi* and *Syracosphaera*
spp. dominated in the latter. Coccolithophore calcite contributed a major fraction of the PIC standing stocks above a depth of
150 m, among which *E. huxleyi* was the most important producer while less abundant and larger species also played a role.
The coccolithophore $CaCO_3$ production rate in the subtropical gyre (0.62 mol m$^{-2}$ yr$^{-1}$) was ~5-fold higher than that in the
Kuroshio-Oyashio transition region (0.14 mol m$^{-2}$ yr$^{-1}$), indicating that inorganic carbon metabolism driven by
coccolithophores is relatively strong in oligotrophic ocean waters. Using a box model including coccolithophore $CaCO_3$
production and metabolic calcite saturation state, we demonstrated that $CaCO_3$ dissolution associated with organic carbon
metabolism can generate excess alkalinity in the oversaturated upper water column of the western North Pacific Ocean. Results
of our study highlight the critical role of coccolithophores in $CaCO_3$ production and dissolution; knowledge of these processes
is important to assess PIC cycling and carbonate pump efficiency in the pelagic ocean.



## 1 Introduction

Calcium carbonate ($CaCO_3$) production and dissolution are two major processes associated with $CaCO_3$ cycling in the ocean,
and a key component of the global oceanic carbon cycle (Broecker and Peng, 1982) via the so-called carbonate pump (Volk
and Hoffert, 1985). Production of biogenic $CaCO_3$ by calcifying plankton in the euphotic zone elevates the partial pressure of
carbon dioxide ($CO_2$) in seawater (e.g., Feely et al., 2002), while ballasting of sinking particles can promote the transport of
carbon from the surface to deep sea and marine sediments (e.g., Armstrong et al., 2001; Klaas and Archer, 2002). Dissolution
of $CaCO_3$ in the water column acts as a buffer to facilitate ocean sequestration of atmospheric $CO_2$ and has the effect of
reducing the rate of ocean acidification (Feely et al., 2004; Barrett et al., 2014). This acidification feedback mechanism makes
it harder for calcifying organisms to produce their skeletons, and thus adversely affects marine ecosystems (Feely et al., 2004;
Steiner et al., 2018). Therefore, quantification of marine $CaCO_3$ production and dissolution is of vital importance in
determining the response of marine ecosystems to changes in the partial pressure of $CO_2$.
Marine $CaCO_3$ occurs in the form of calcite, aragonite and high-magnesium calcite. Coccolithophores are a key, single-
celled phytoplankton taxonomic group, responsible for a large percentage (30–60 %) of modern oceanic $CaCO_3$ production
and 10–20 % of marine primary production on a global scale (Poulton et al., 2006, 2013). Coccolithophore calcite accounts
for a major fraction of the $CaCO_3$ exported to the deep sea and sediments (Broerse et al., 2000; Young and Ziveri, 2000; Rigual
Hernández et al., 2020). Data assessment along a northeast Pacific transect from Hawaii to Alaska suggested that
coccolithophore calcite comprises 90 % of the total $CaCO_3$ production in the euphotic zone, while pteropods and foraminifera
only play a minor role (Ziveri et al., 2023). However, large uncertainties remain in estimates of the production rate of $CaCO_3$
in the upper ocean (Balch et al., 2007; Berelson et al., 2007; Smith and Mackenzie, 2016; Ziveri et al., 2023). Based on a
global compilation of $CaCO_3$ production using in situ $^{14}C$ incubations, Daniels et al. (2018) found that calcification rate ranged
from 10 to 600 mg m$^{-2}$ d$^{-1}$. A recent estimate of $CaCO_3$ biomass from three main pelagic calcifying plankton groups also
suggested large variation in $CaCO_3$ production in the eastern North Pacific Ocean, ranging from 110 to 729 mg m$^{-2}$ d$^{-1}$ (Ziveri
et al., 2023). The generally low coverage of observations and the considerable spatiotemporal variation shown by available
data, along with the current scarcity of studies, most likely result in deviations among regional estimates.
$CaCO_3$ dissolution is generally assumed to mainly occur below the saturation horizon. However, this assumption has been



challenged by an increasing number of studies which suggest considerable dissolution in the oversaturated upper water-column
where $CaCO_3$ saturation state ($\Omega$) is greater than 1 (Sabine et al., 2002; Chung et al., 2003; Berelson et al., 2007; Barrett et al.,
2014). Shallow-water $CaCO_3$ dissolution is supported by excess alkalinity/calcium (Feely et al., 2002, 2004; Cao and Dai,
2011; Subhas et al., 2022), decreased sinking fluxes of particulate inorganic carbon (PIC) measured using the $^{234}$Th method,
and sediment trap data form the upper ocean (Dong et al., 2019; Roca-Martí et al., 2021). One possible mechanism for $CaCO_3$
dissolution in calcite-oversaturated shallow waters is the dissolution of more soluble forms of $CaCO_3$ including aragonitic
pteropods and high-Mg fish calcites (Honjo et al., 2008; Wilson et al., 2009; Dong et al., 2019; Folkerts et al., 2024; Oehlert
et al., 2024). An alternative hypothesis is that microbial oxidation of organic matter produces an acidic microenvironment
conducive for carbonate dissolution (Bishop et al., 1980; Milliman et al., 1999). The digestive system of grazing zooplankton
may also contain acidic conditions facilitating $CaCO_3$ dissolution (Pond et al., 1995; White et al., 2018).
The North Pacific Ocean is a vital region for modulating the carbon cycle, as it accounts for ~25 % of the global ocean sink
for atmospheric $CO_2$ (Takahashi et al., 2009). In the eastern North Pacific Ocean, $CaCO_3$ production, export, and dissolution
have been studied along a transect from Hawaii to Alaska (Dong et al., 2019, 2022; Naviaux et al., 2019; Subhas et al., 2022;
Ziveri et al., 2023), which revealed that depth-integrated $CaCO_3$ production in the nutrient-rich subpolar gyre is twice as high
as that in the nutrient-poor subtropical gyre. This contrast, however, is smaller than the sixfold to sevenfold difference based
on satellite estimates of surface PIC, indicating the importance of coccolithophore $CaCO_3$ production in deep waters. These
investigations also suggested that widespread shallow-water $CaCO_3$ dissolution is driven by metabolic activity along the
Hawaii-to-Alaska transect.
Here, we determined the abundances and species compositions of coccolithophore, as well as the concentrations of
coccolithophore calcite and PIC based on both Niskin bottle and in situ pump sampling in the upper water column of the
western North Pacific Ocean. Additionally, we conducted measurements of environmental conditions such as nutrient and
carbonate chemistry parameters. The aims of this research were to answer the following questions: (1) What is the distribution
of coccolithophore abundances and species compositions across the oligotrophic-nutrient replete environmental gradient? (2)
What is the contribution of coccolithophores to $CaCO_3$ production in the euphotic zone? (3) Does shallow-water $CaCO_3$
dissolution occur in the western North Pacific Ocean, and what is the significance of metabolic activities in driving dissolution



in oversaturated ambient conditions?

## 2 Methods

### 2.1 Sample collection

Sampling was conducted onboard R/V *Tan Kah Kee* during cruise NORC2022-306 from 09 June to 25 July 2022. The cruise
trajectory crossed from the oligotrophic North Pacific Subtropical Gyre (NPSG) to the relatively nutrient-rich Kuroshio-
Oyashio transition region along the 155°E meridian (Fig. 1a; Table S1). Seven sampling stations can be divided into those
located in the NPSG region, including stations M30, M32 and M35, characterized by high sea-surface temperature (SST) and
low surface chlorophyll *a* (Chl *a*) and PIC concentrations, and those located in the Kuroshio-Oyashio transition region,
including stations KE3, STN41, STN43 and STN45, featuring lower SST, but higher Chl *a* and PIC concentrations (Fig. 1b–
d).

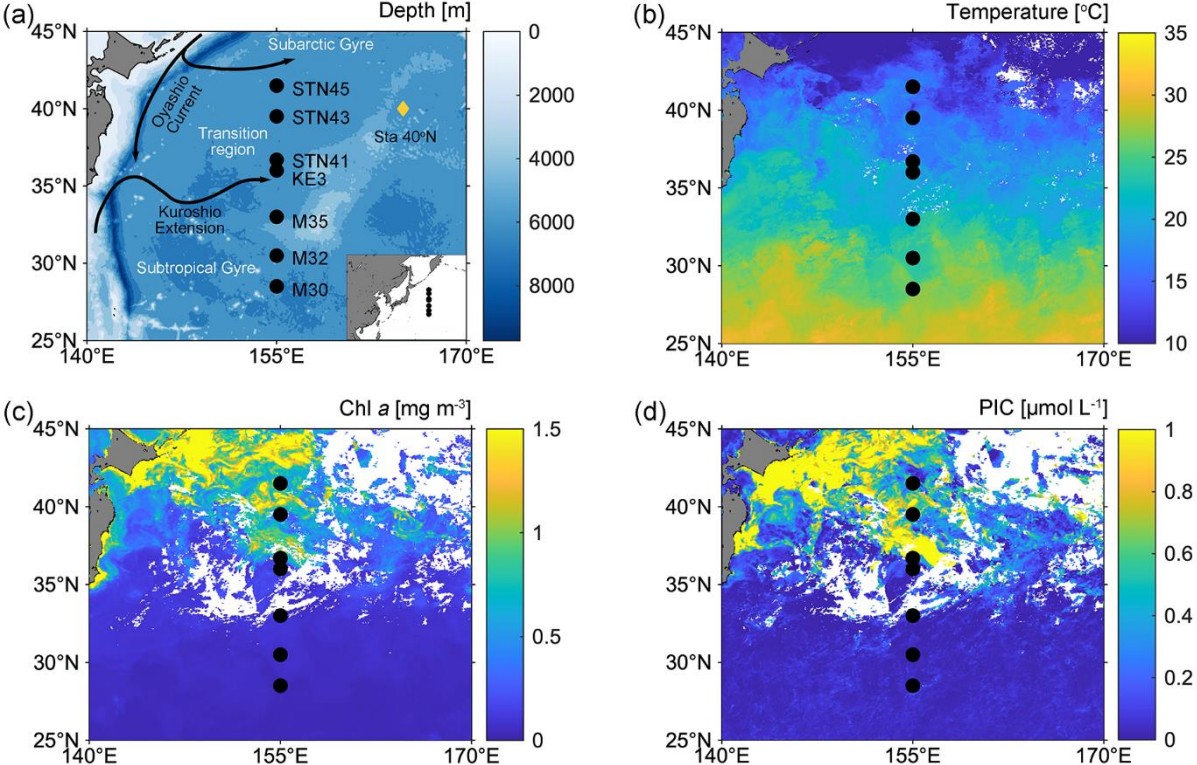

**Fig 1.** (a) Map of the western North Pacific Ocean showing sampling stations (black filled circles) and major surface currents (solid black lines). The yellow diamond indicates station 40° N at which particulate inorganic carbon (PIC) sinking flux was investigated by Honda et al., (2002); (b–d) satellite-based temperature, chlorophyll *a* (Chl *a*) and particulate inorganic carbon (PIC) concentrations in surface water in June 2022 (data from the Moderate Resolution Imaging Spectroradiometer (MODIS)-Aqua satellite; https://oceancolor.gsfc.nasa.gov/l3/).

Water samples were collected within the water column above 300 m depth using Niskin bottles on a rosette system equipped with SBE-911 conductivity-temperature-depth (CTD) sensors (Sea-Bird Electronics, Inc., Bellevue, WA, USA). For PIC analyses, 24 L of seawater were collected using acid-cleaned fluorinated bottles and filtered through two quartz microfiber (QMA) filters (1.0 μm pore size, 25 mm diameter). For coccolithophore analyses, 2–4 L of seawater were collected and gently filtered through polycarbonate membranes (0.8 μm pore size, 25 mm diameter), using a vacuum pump at <20 mm Hg pressure.



Filters were oven-dried at 60°C and stored in plastic petri dishes.
Size-fractionated particles were collected using McLane Research in situ pumps. Filter holders were loaded with a 51 μm
Sefar polyester mesh prefilter followed by paired Whatman QMA filters. Hereafter, we refer to the two particle size fractions
as large and small size fractions (LSF, > 51 μm and SSF, 1–51 μm, respectively). A 1/4 subsample of the 51 μm polyester mesh
prefilter and two circles of 23 mm diameter subsample of the QMA filter were analyzed for LSF and SSF PIC concentrations,
respectively, and the sum of the two fractions yielded the total PIC ($PIC_{total}$) concentration.
**2.2 Sample analyses**
PIC concentrations were determined by measuring the amount of $CO_2$ released after acid treatment of the filters using a Thermo
Delta V Plus isotope ratio mass spectrometer (IRMS, Thermo Fisher, USA) coupled with a Thermo Gasbench II system at the
Center for Isotope Geochemistry and Geochronology of the Laoshan Laboratory (Li et al., 2021). International reference
materials of calcite NBS-18 and IAEA-603 were measured for calibration. The PIC analytical precision was better than 10 %
(one standard deviation, 1SD).
A portion of the filters was mounted with a carbon sticky tab on a stub and gold-coated prior to analysis using a Quanta 650
FEG field-emission scanning electron microscope (SEM). The coccosphere cell or detached coccolith concentrations (CC,
cells or coccoliths $L^{-1}$) were estimated as follows:
$CC = (F * C)/(V * S)$    (1)
where F is the effective filtration area (336.9 $mm^2$), C is the total number of coccosphere cells or detached coccoliths, V is
the filtered seawater volume, and S is the total area of fields of view ($mm^2$). This cell counting strategy gives a detection limit
of at least 1.87 cells $mL^{-1}$ (Bollmann et al., 2002). Coccolithophore species identification followed Young et al. (2003) and the
Nannotax3 website (http://ina.tmsoc.org/Nannotax3/). Individual coccolithophore calcite content was calculated by
multiplying the number of coccoliths per cell by the average coccolith calcite mass of a given species (Young and Ziveri, 2000;
Yang and Wei, 2003; Boeckel and Baumann, 2008; Beuvier et al., 2019; Jin et al., 2022). All the biometry work based on SEM
images was conducted using imageJ free software (imagej.nih.gov/ij/) and Coccobiom2-SEM measuring macro (Young, 2015).
**2.3 Estimation of CaCO₃ production rate**
CaCO₃ production rates above 150 m were determined by dividing measurements of the living CaCO₃ standing stock (which





only included whole coccosphere cells and excluded loose coccoliths) by the coccolithophore turnover time, which is 0.7–10
days with a growth rate ranging from 0.1 to 1.5 cell divisions day$^{-1}$ (Krumhardt et al., 2017; Ziveri et al., 2023). Uncertainty
in the CaCO$_3$ standing stock estimates, which were obtained by vertically integrating PIC concentrations above a depth of 150
m, was typically ±10 % (1SD).
A Monte Carlo-based probabilistic approach was used to determine the CaCO$_3$ production rate and the uncertainties
associated with the turnover time using a flat probability distribution in MATLAB visualized by a violin plot (Hoffmann, 2015).
To obtain an annual CaCO$_3$ production based on our field observations, we used the ratio of satellite-derived PIC for July 2022
to annual climatology PIC (data from the NASA Goddard Space Flight Center's Ocean Ecology Laboratory) to calibrate for
potential seasonal variability (Ziveri et al., 2023).
**2.4 Influence of environmental conditions on coccolithophores**
The redundancy analysis (RDA) function in the vegan package in R, in combination with Monte Carlo permutation, were
performed to assess the relative importance of environmental variables in explaining the overall variation in the composition
of the coccolithophore community (Oksanen et al., 2007). The contribution of each environmental variable to community
variation was determined by hierarchical partitioning in canonical analysis via the 'dbRDA' function in the "rdacca.hp"
package in R (Lai et al., 2022). We further conducted random forest analyses to identify the main predictors of coccolithophore
abundance with the "randomForest" package in R (Liaw and Wiener, 2002).
**2.5 Evaluation of shallow-water CaCO$_3$ dissolution**
To estimate the effect of microenvironment undersaturation driven by microbial oxidation of organic matter, we assumed that
aerobic metabolic activity of marine particles consumes all ambient oxygen and alters ambient dissolved inorganic carbon
(DIC) and total alkalinity (TA). This process decreases microenvironment $\Omega$ and thus causes CaCO$_3$ dissolution. We calculated
$\Omega$ due to in situ metabolism (defined as $\Omega_{met}$) using the concurrently altered DIC, TA and soluble reactive phosphate (SRP)
concentrations following Subhas et al., (2022):
$DIC_{met} = DIC + [O_2] * R_{CO}$  (2a)
$TA_{met} = TA - [O_2] * R_{NO}$  (2b)
$SRP_{met} = SRP + [O_2] * R_{PO}$  (2c)



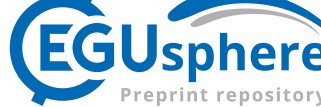

where $R_{CO}$, $R_{NO}$, and $R_{PO}$ denote the Redfield ratio of carbon to oxygen (0.688), nitrate to oxygen (0.0941), and phosphate to
oxygen (0.0059), respectively (Anderson and Sarmiento, 1994).
A one-dimensional (1-D) model was used to diagnose the sinking flux and $CaCO_3$ dissolution rate in the upper 1000 m of
the water-column by coccolithophore $CaCO_3$ production, coccolith dissolution kinetics and $\Omega_{met}$ (Dong et al., 2019; Subhas et
al., 2022). The model assumes that dissolution of $CaCO_3$ particles occurs when the water column $\Omega_{met} < 1$. The magnitude of
dissolution thus depends on the initial sinking flux, $\Omega_{met}$ and particle residence time as follows:
$$Flux_{zi} = Flux_{zi-1} * \left(1 - R_{diss} * \frac{Z_i - Z_{i-1}}{w}\right) \quad (3)$$
where $Flux_{zi}$ and $Flux_{zi-1}$ (mmol m$^{-2}$ d$^{-1}$) denote the $CaCO_3$ sinking flux at a given depth $Z_i$ and its overlying depth $Z_{i-1}$,
respectively, $R_{diss}$ (g g$^{-1}$ d$^{-1}$) denotes $CaCO_3$ dissolution rate which is a function of the $\Omega_{met}$ at depth $Z_{i-1}$ (Subhas et al., 2018;
Subhas et al., 2022) and w (m d$^{-1}$) denotes the particle sinking rate.
TA regeneration rate ($R_{TA}$, µmol kg$^{-1}$ yr$^{-1}$) at depth $Z_i$ resulting from shallow-water $CaCO_3$ dissolution was calculated based
on the following equation:
$$R_{TA\_zi} = \frac{2 * Flux_{zi} * R_{diss}}{\rho * w} \quad (4)$$
where $\rho$ denotes the density of seawater (1029 kg m$^{-3}$).

**3 Results**
**3.1 Hydrography and nutricline**
Temperature and salinity were high in surface water and gradually decreased with increasing depth above 300 m.
Hydrochemical variables clearly exhibited a south to north trend. Temperature and salinity were highest at the surface of station
M30, due to strong net evaporation in the subtropical gyre (Fig. 2a and b). There was a northward decrease in temperature and
salinity due to the influence of upwelling in the subarctic gyre. The surface mixed layer depth, defined as the depth where
potential density increases by 0.03 kg m$^{-3}$ compared to that at the sea surface, varied around 11–25 m.
In contrast to temperature and salinity and as expected, the distribution of dissolved inorganic nitrogen (DIN, nitrate plus
nitrite), SRP and dissolved silicate (DSi) showed a generally northward increasing pattern (Figs. 2c–d and S1). Surface DIN



concentrations were below the detection limit in the NPSG region and averaged 0.02 µmol L$^{-1}$ in the Kuroshio-Oyashio
transition region. The top of the nutricline, defined as the depth where DIN concentrations reach 0.1 µmol L$^{-1}$, ranged from
110 m at station M30 to 20 m at station STN45. The ammonium (NH$_4^+$) concentration above 100 m at station STN45 was
notably higher than that at other stations (Fig. S1b). Similar to the nutricline distribution, the deep chlorophyll maximum
(DCM) depth gradually shoaled northward from 110 m at station M30 in the NPSG region to 33 m at station STN45 in the
Kuroshio-Oyashio transition region (Fig. 2e).






**Fig. 2.** Vertical depth distributions of (a) temperature, (b) salinity and concentrations of (c) soluble reactive phosphate (SRP),
(d) dissolved inorganic nitrogen (DIN, nitrate plus nitrite), (e) Chlorophyll *a* (Chl *a*), (f) particulate inorganic carbon (PIC),
(g) coccosphere cell and (h) detached coccolith in the upper 300 m of the water column in the study area.

### 3.2 Vertical distribution of PIC and coccolithophore concentrations

PIC concentrations along the 155°E transect ranged from 0.02 to 0.17 µmol $L^{-1}$, with an average of 0.06 ± 0.04 µmol $L^{-1}$ in
the upper 300 m of the water column (Fig. 2f). Generally, PIC concentrations were lower at the surface and increased with
increasing depth to attain a maximum in the DCM layer, and decreased with depth thereafter. In the DCM layer, PIC
concentrations ranged from 0.06 µmol $L^{-1}$ at 110 m of station M30 in the subtropical gyre to 0.16 µmol $L^{-1}$ at 33 m of station
STN45 in the Kuroshio-Oyashio transition region. The vertical distribution pattern of bottle-derived PIC and coccosphere cell
concentrations overall followed that of Chl *a*, showing a northward shoaling of the subsurface maximum.

Concentrations of coccosphere cells ranged from ca. 970 to 75,000 cells $L^{-1}$ (Fig. 2g). Along the transect, a subsurface
maximum was evidenced around the DCM layer with an average of 42,000 cells $L^{-1}$, followed by a steep decrease below 100
m. The highest coccosphere cell concentration was observed at 65 m of station STN41, corresponding to the highest PIC
concentration. The average coccosphere cell concentration was notably lower in the NPSG region (9,800 cells $L^{-1}$) than in the
transition region (18,000 cells $L^{-1}$). The detached coccolith concentration averaged 340,000 coccoliths $L^{-1}$, with a range of
11,000 to 800,000 coccoliths $L^{-1}$ (Fig. 2h). The highest concentration was observed around 10–40 m of station STN43. High
coccolith concentrations were also observed below 100 m at stations M32, M35 and STN41.

Size-fractionated PIC concentrations from in situ pumps varied from 0.01 to 0.09 µmol $L^{-1}$ in the SSF and from 0.01 to 0.06
µmol $L^{-1}$ in the LSF. PIC$_{total}$ concentrations averaged 0.07 ± 0.02 µmol $L^{-1}$, and were comparable to bottle-derived PIC
concentrations (Fig. 3). Roughly 70 % of the PIC was contributed by the SSF at each sampling station. Generally, LSF PIC
concentrations increased northward from stations M30–M35 to stations KE3–STN45 and accounted for 22 % and 36 % of
PIC$_{total}$ concentrations in the NPSG region and the Kuroshio-Oyashio transition region, respectively. The maximum
concentration of LSF PIC (0.06 µmol $L^{-1}$) was observed at 26 m of station STN45 (Fig. 3g).



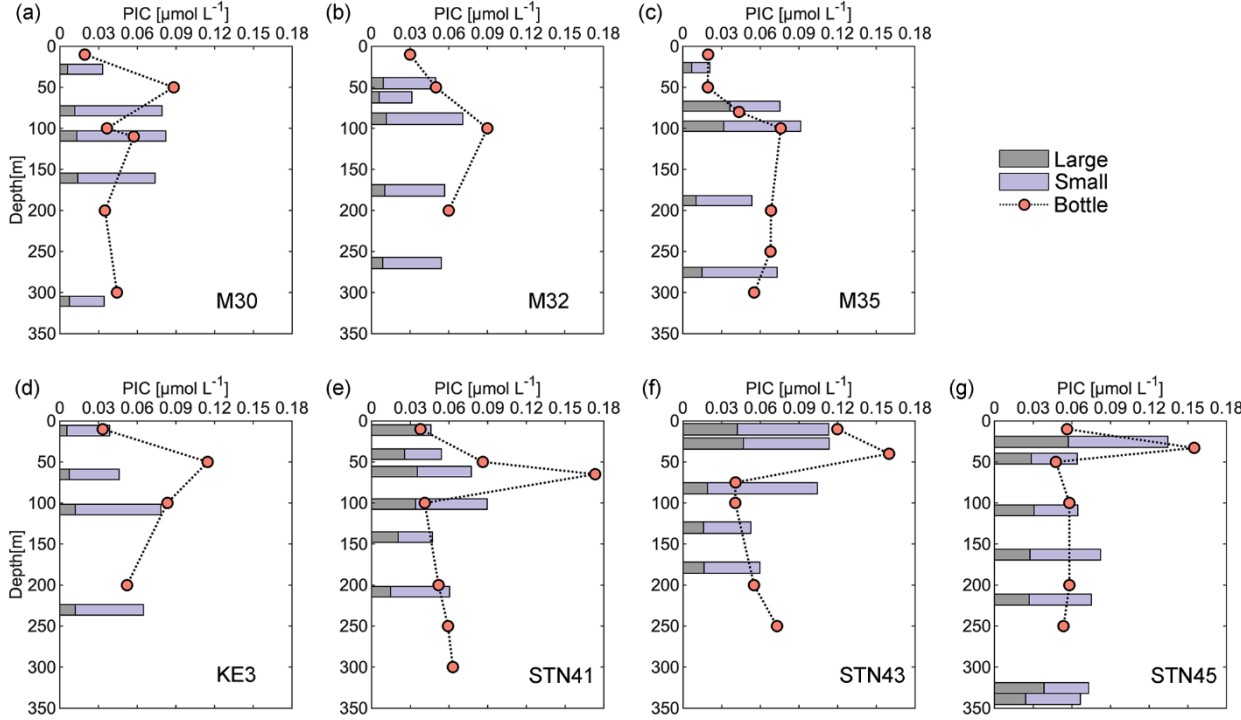

**Fig. 3.** Vertical depth distributions of particulate inorganic carbon (PIC) concentrations derived from sampling using both Niskin bottles and in situ pumps (small size fraction of 1–51 μm and large size fraction of > 51 μm) in the upper 350 m of the water column at sampling stations in the study area.

## 3.3 Characteristics of the coccolithophore assemblage

Coccolithophore populations were predominantly represented by *Emiliania huxleyi*, *Gephyrocapsa ericsonii*, *Gephyrocapsa oceanica*, *Umbellosphaera tenuis*, *Syracosphaera* spp., holo-coccolithophores (HOL), *Algirosphaera robusta,* and *Florisphaera profunda* (comprising > 1 % of total coccosphere abundance; Fig. 4). In surface water, coccolithophore cells were dominated by *Dicosphaera tubifera*, *U. tenuis* and HOL at stations M30 and M32 (Fig. 4a and b) and by *G. ericsonii* at stations M35, KE3 and STN41 (Fig. 4c, d and e), while high abundance of *E. huxleyi* and *Syracosphaera* spp. was clearly observed at stations STN43 and STN45 (Fig. 4f and g). It is noteworthy that *E. huxleyi* contributed the largest fraction (50 %) to the total coccolithophore assemblage and was also found to be the dominant species in the DCM layer. *U. tenuis* was mainly





observed in subtropical gyre waters, with peak abundance at 50 m and lower abundance at the surface and in DCM waters (Fig. 4a and b). Lower euphotic zone (LPZ) coccolithophore species (including *A. robusta* and *F. profunda*) were commonly found in the subsurface population below 50 m, accounting for 7 % of the entire coccolithophore community. Overall, coccolithophores were scarce in the NPSG region and dominated by *U. tenuis*, whereas their abundance notably increased in the Kuroshio-Oyashio transition region where it was dominated by *E. huxleyi*, *Gephyrocapsa* and *Syracosphaera* spp..

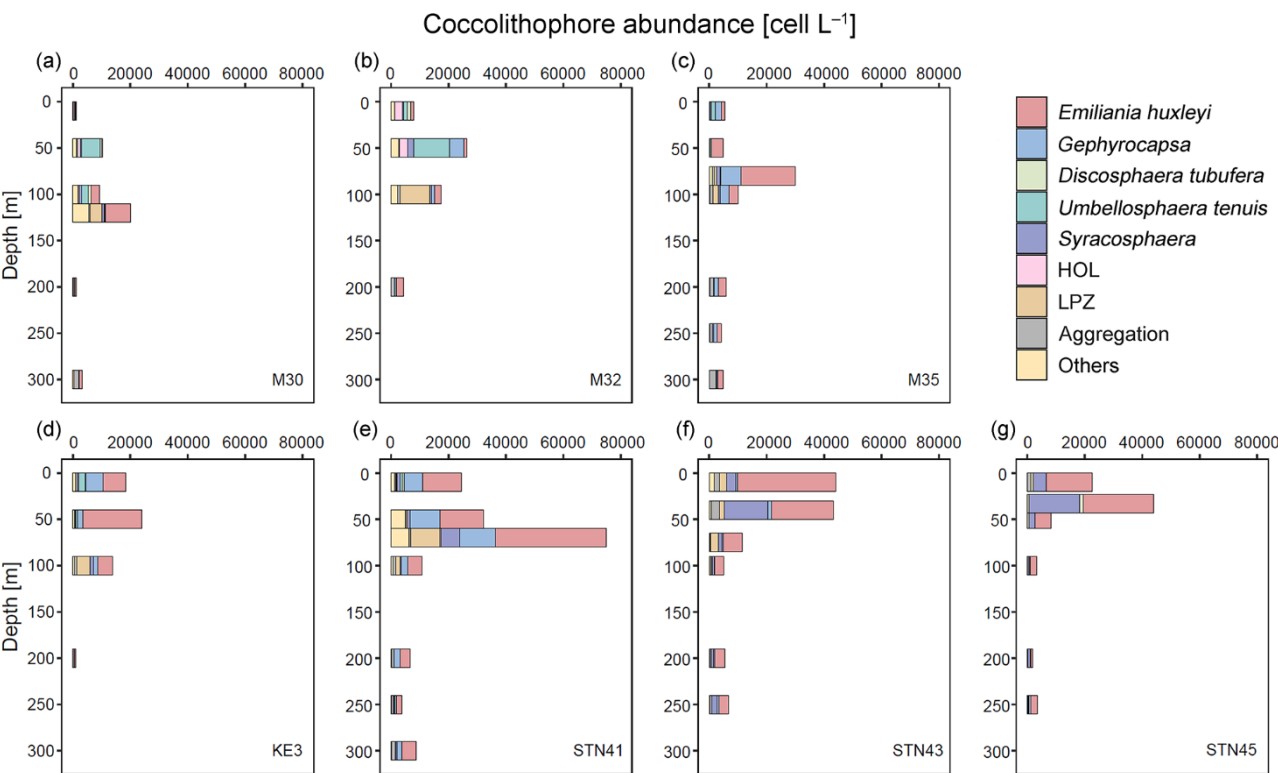

**Fig. 4.** Abundance of different coccolithophore groups in the upper 300 m of the water column at sampling stations in the study area. Lower euphotic zone (LPZ) species include *Florisphaera profunda* and *Algirosphaera robusta*; HOL indicates holo-coccolithophores.

The estimated coccolithophore calcite concentrations ranged from 0.00 to 0.23 μmol L$^{-1}$ averaging 0.05 ± 0.04 μmol L$^{-1}$

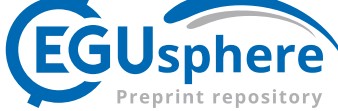

above 300 m along the 155°E transect. *E. huxleyi* accounted for 50 % of the total coccolithophore assemblage but represented
only 9 % of the coccolithophore calcite concentration (Fig. 5a and c). The less abundant (<1 %) species *Calcidiscus leptoporus*
and *Oolithotus fragilis* accounted for 7.5 % and 7.7 % of the coccolithophore calcite concentration, respectively.
*Syracosphaera* spp. was the largest contributor, accounting for 17.7 % of the coccolithophore calcite concentration (Fig. 5c).
Additionally, *E. huxleyi* detached coccoliths, comprising 94 % of the total detached coccoliths, made up ~16 % of the total
coccolithophore calcite concentration (Fig. 5b and c).

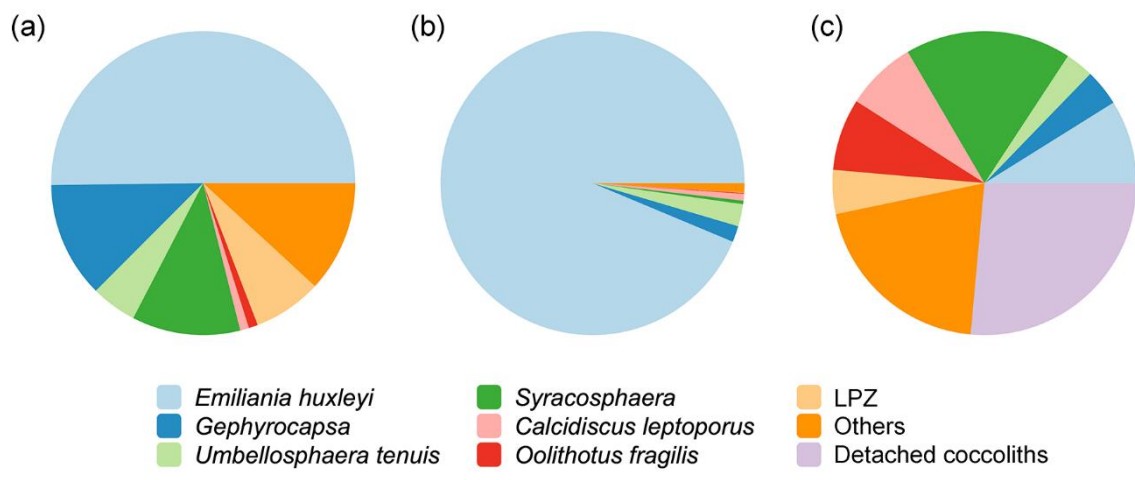


**Fig. 5.** Contribution of different coccolithophore groups to (a) coccosphere cell abundance, (b) detached coccolith abundance
and (c) coccolithophore calcite concentration in the upper 300 m of the water column. Lower euphotic zone (LPZ) species
include *Florisphaera profunda* and *Algirosphaera robusta*.

**3.4 CaCO$_3$ standing stock and production**
The standing stock of CaCO$_3$ was determined by considering the first shallow sampling depth to a consistent depth of 150 m
(Figs. 6a and S2a). CaCO$_3$ standing stock derived from Niskin bottle-sampling ranged from 660 to 1,200 mg m$^{-2}$, and was
lower in the oligotrophic NPSG region (790 mg m$^{-2}$) than in the relatively nutrient-high Kuroshio-Oyashio transition region
(1,100 mg m$^{-2}$). Based on the estimated coccolithophore calcite concentrations, CaCO$_3$ standing stocks ranged from 370 to



1,000 mg m$^{-2}$ and peaked at station STN41. Calcite from coccolithophores comprised on average 76 % of the CaCO$_3$ standing
stock from Niskin bottle samples, and the contribution was higher in the NPSG region (91 %) than in the Kuroshio-Oyashio
transition region (65 %) (Fig. 6a), demonstrating the vital role of coccolithophores in CaCO$_3$ production, particularly in
oligotrophic ocean waters.

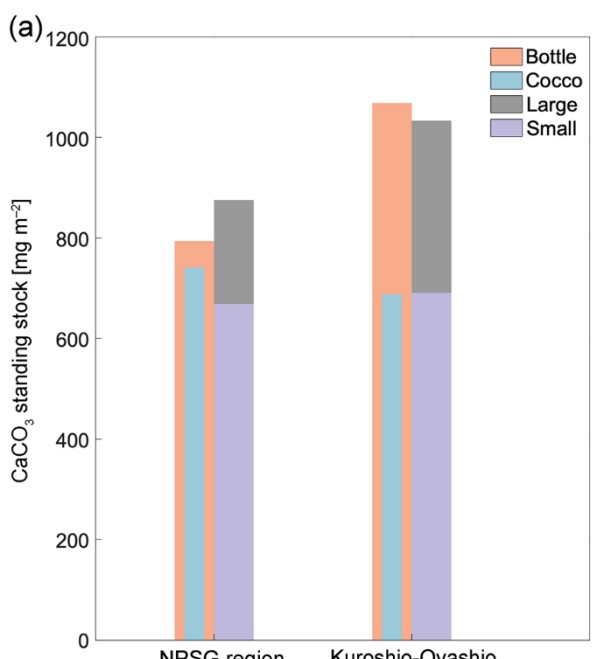
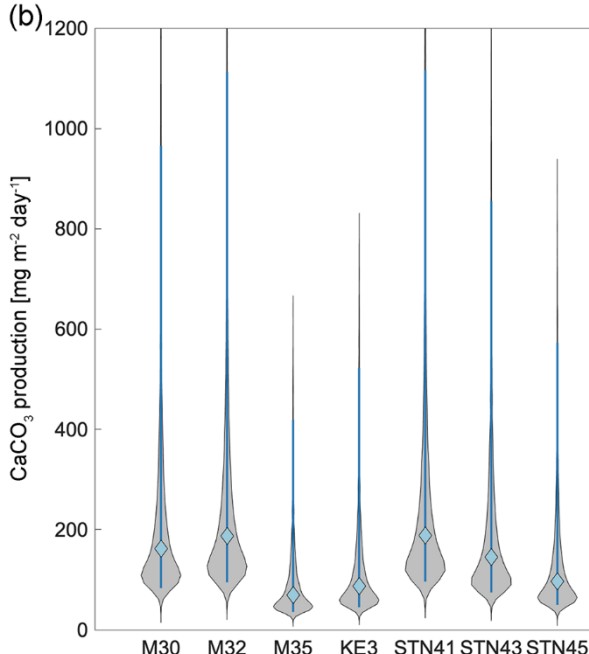


**Fig. 6.** (a) Calcium carbonate (CaCO$_3$) standing stock in the upper 150 m water column estimated from Niskin bottle particulate
inorganic carbon (PIC), total calcite (Cocco) and size-fractionated (large and small fractions indicate > 51 and 1–51 μm,
respectively) PIC concentrations in the North Pacific Subtropical Gyre (NPSG) and Kuroshio-Oyashio transition regions; (b)
CaCO$_3$ production by coccolithophores in the upper 150 m water column at indicated sampling stations in June-July 2022.

Total CaCO$_3$ standing stock derived from in situ pump samples ranged from 820 to 1,300 mg m$^{-2}$, averaging 880 mg m$^{-2}$ in
the subtropical gyre and 1,030 mg m$^{-2}$ in the transition region, consistent with results from Niskin bottle samples. The CaCO$_3$
standing stock of the SSF ranged from 514 to 904 mg m$^{-2}$ and accounted for 71 % of the total standing stock in the entire





research domain (Fig. 6a).
Given that coccolithophores have a turnover time of 0.7–10 days (Krumhardt et al., 2017; Ziveri et al., 2023), CaCO$_3$
production rate in the upper 150 m of the water column ranged from 70 to 190 mg m$^{-2}$ d$^{-1}$ during the sampling period (Fig.
6b). Generally, the coccolithophore CaCO$_3$ production was comparable in the subtropical gyre and the Kuroshio-Oyashio
transition region, averaging $140 \pm 62$ and $130 \pm 46$ mg m$^{-2}$ d$^{-1}$, respectively. Coccolithophore CaCO$_3$ production was maximal
at station STN41 where it reached 190 mg m$^{-2}$ d$^{-1}$, corresponding to the maximum coccosphere cell concentration (Fig. 2e).
The lowest coccolithophore CaCO$_3$ production of 70 mg m$^{-2}$ d$^{-1}$ was observed at M35 in the NPSG region.

**4 Discussion**
**4.1 Coccolithophore responses to environmental factors**
Coccolithophores are an important component of phytoplankton biomass and fill a variety of ecological niches in global oceans.
They inhabit different marine environments from oligotrophic to eutrophic, warm to cold, euphotic to aphotic, and stratified
to mixed (Balch, 2018). It is of critical importance to evaluate the response of coccolithophore species to different
environmental conditions to better understand changes in coccolithophore species diversity, community composition, and their
role in the oceanic carbon cycle.
Although biogeographical zones of coccolithophores in the North and Central Pacific were identified a couple of decades
ago, few studies have investigated coccolithophore distributions in the North Pacific over the recent two decades (Okada and
Honjo, 1973; Hagino et al., 2005). In the western North Pacific Ocean, higher diversity and less abundant coccolithophore
assemblages were observed in the oligotrophic subtropical gyres, whereas the Kuroshio-Oyashio transition region tended to
exhibit a lower diversity corresponding to higher PIC and coccolithophore concentrations (Figs. 2 and S3). This finding is
consistent with results from the Atlantic Ocean, and a result of the different survival strategies of various coccolithophore
species (Balch et al., 2019). Coccolithophores are nutrient stress tolerant and have lower iron cell quotas, and are thus generally
abundant in the open ocean (Gregg and Casey, 2007; Brun et al., 2015). When nutrients and light are plentiful, the heavy
coccoliths of this group of phytoplankters pose a selective disadvantage over diatoms and chlorophytes (Gregg and Casey,
2007). The majority of coccolithophore species are K-selected, characterized by relatively slow-growth, large cell size and are



more competitive in low-nutrient and well-stratified regions (Brand, 1994), whereas only few r-selected species (such as the
fast-growing and small-sized *E. huxleyi*) mainly survive in relatively dynamic and nutrient-rich regions (Charalampopoulou,
2011; Brun et al., 2015; O'brien et al., 2016). In the present study, the most abundant and widely distributed coccolithophore
species was *E. huxleyi* which showed increasing abundance northward along the study transect (Fig. 4). This is consistent with
prior observations demonstrating that *E. huxleyi* is the most abundant coccolithophore species in the subarctic, subantarctic
and bordering transitional regions (Saavedra-Pellitero et al., 2014).
According to RDA results, environmental variables accounted for 47.6 % of the total variation in coccolithophore
community composition (Fig. 7a). The first two RDA axes suggested that there were significant spatial differences in the
coccolithophore community across depths and regions (Fig. S4). Correspondingly, hierarchical partitioning analysis showed
that depth and latitude had a significant effect on coccolithophore community variation ($p < 0.05$). Other environmental factors,
such as temperature, salinity, Chl *a* and TA were also important influencing the coccolithophore community (Fig. 7b).
Based on Spearman's correlation analysis, coccolithophore abundance showed a significant positive relationship with
temperature, $\Omega_{calcite}$ and pH, and a significant negative relationship with depth, DIC and macro-nutrient concentrations,
especially for *D. tubifera*, *U. tenuis* and HOL that are more sensitive to environmental factors (Fig. 7c). The positive correlation
with temperature is consistent with field observations and model simulations pointing to a general trend of increasing
coccolithophore abundance in the context of global warming (Rivero-Calle et al., 2015; Rousseaux and Gregg, 2015). More
abundant species like *E. huxleyi* and *Syracosphaera* spp., however, only showed a highly positive correlation with depth,
latitude and Chl *a* concentration, suggesting that these species are more adaptable to varying environmental conditions
(Schlüter et al., 2014). Using random forest analysis, we also determined that the best predictors of coccolithophore abundance
were Chl *a* concentration, depth and $\Omega_{calcite}$ ($p < 0.01$; Fig. 7d).



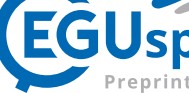

**Fig. 7.** (a) Redundancy analysis (RDA) diagram illustrating the relationship between the coccolithophore community and environmental factors; (b) independent contribution of each environmental factor to coccolithophore community variation using hierarchical partitioning-based canonical analysis; (c) correlations between coccolithophore groups and environmental factors with color gradients denoting the significance of the Spearman's correlation coefficient r. Asterisks represent the statistical significance (***$p < 0.001$, **$p < 0.01$, *$p < 0.05$); (d) random forest mean predictor importance, i.e., the percentage of increase in the mean variance error (MSE) of environmental factors on coccolithophore abundance (**$p < 0.01$, *$p < 0.05$, n.s., non-significant). Chl $a$: chlorophyll $a$, DIC: dissolved inorganic carbon, TA: total alkalinity, $\Omega_{calcite}$: saturation state with respect to calcite, PIC: particulate inorganic carbon, DIN: dissolved inorganic nitrogen (nitrate plus nitrite), $NH_4^+$: ammonium,



SRP: soluble reactive phosphate, DSi: dissolved silicate, HOL: holo-coccolithophores and LPZ: lower euphotic zone species
*Florisphaera profunda* and *Algirosphaera robusta*.

**4.2 Contribution of coccolithophore calcite to PIC**

In this study, bottle- and pump-derived PIC concentrations generally agreed with each other (Fig. 3), and both were comparable
to and of the same order of magnitude as the suspended PIC concentrations detected in the Atlantic, Indian and Pacific Oceans
(Beaufort et al., 2008; Barrett et al., 2014; Lam et al., 2015, 2018; Maranón et al., 2016). Coccolithophore calcite
concentrations showed a significant positive correlation with PIC concentrations ($r^2 = 0.75$, $p < 0.01$, n = 40; Fig. 8a),
highlighting the major contribution of coccospheres and detached coccoliths (68 %) to the total $CaCO_3$ in the upper 300 m of
the water column. This is consistent with findings from the eastern North Pacific Ocean where coccolithophores dominate
$CaCO_3$ production (Ziveri et al., 2023). It is noteworthy that detached coccolith concentrations of *E. huxleyi, U. tenuis* and
*Syracosphaera* spp. showed a significant positive relationship with their coccosphere cell concentrations (Fig. 8b–d),
indicating that those detached particles were likely to have originated from living cells.

**Fig. 8.** Relationship of (a) coccolithophore calcite vs particulate inorganic carbon (PIC) concentrations and (b–d) detached coccolith vs coccosphere cell concentrations for (b) *Emiliania huxleyi*, (c) *Umbellosphaera tenuis* and (d) *Syracosphaera* spp. in the upper 300 m water column at the study site. Equations describing the fitted straight lines are also shown.

Abundant coccolithophore groups, including *E. huxleyi*, *Syracosphaera* spp., LPZ species and aggregation, showed a significant positive relationship with PIC concentration (Fig. 7c), but less abundant species like *C. leptoporus* and *O. fragilis* also made a large contribution to calcite concentrations. It has been reported that despite the relatively low abundance (< 2 %)



of the coccolithophore community, some larger species such as *C. leptoporus*, *Helicosphaera carteri* and *Coccolithus pelagicus*
could account for most of the coccolithophore CaCO₃ flux to the deep ocean (Rigual Hernández et al., 2020). Some rare
coccolithophore species with high coccolith and coccosphere cell concentrations have also been identified as important
contributors to both upper-ocean calcite production (Daniels et al., 2016) and deep-sea calcite fluxes (Ziveri et al., 2007). Thus,
although *E. huxleyi* is one of the most abundant species in the ocean, larger coccolithophore species can also play an important
role in CaCO₃ export.
Higher CaCO₃ standing stock in the upper 150 m of the Kuroshio-Oyashio transition region (Fig. 6a) is consistent with
satellite observations suggesting that higher surface PIC concentrations occur at high latitudes (Balch et al., 2005; Berelson et
al., 2007). In the present study, however, the relative contribution of coccolithophores to the CaCO₃ standing stock was higher
in the NPSG region (~91 %) than in the Kuroshio-Oyashio transition region (~65 %) (Fig. 6a). To date, most studies estimated
CaCO₃ standing stock using satellite-derived data, which might be challenging to use in subtropical gyres where the DCM
depth usually lies below 100 m. Because coccolithophore CaCO₃ is largely produced in the lower layer of the euphotic zone
and is thus difficult to detect by satellites, coccolithophore contributions could be underestimated.
In oligotrophic ocean gyres, subsurface CaCO₃ production could still occur even if surface PIC is low (Balch et al., 2018).
Along our studied transect, maximum coccolithophore abundances increased about twofold from the subtropical gyre to the
transition region (Fig. 2e), while a much smaller difference was found in the integrated coccolithophore CaCO₃ between the
two regions (Fig. 6a). This suggests that subsurface coccolithophore CaCO₃ contributed substantially to the total upper water
column PIC flux in the NPSG. Coccolithophore groups were diverse in the subtropical gyre, where the environmental
conditions favor slow-growing, large and heavy species, which account for a large fraction of CaCO₃ production. In the
Southern Ocean, coccolithophores contribute to the highest annual CaCO₃ export in waters with low algal biomass
accumulations (Rigual Hernández et al., 2020). Given that low surface PIC regions (< 0.1 mmol m⁻³) occupy ~87 % of the
global ocean surface (Ziveri et al., 2023), our data suggest that coccolithophore CaCO₃ production in subsurface waters
constitutes an important part of global pelagic CaCO₃ production.
Size-fractionated PIC concentrations showed a smaller contribution of coccolithophores to the CaCO₃ standing stock in the
Kuroshio-Oyashio transition region (67 %) than in the NPSG region (76 %) (Fig. 6a), consistent with observations from the



eastern North Pacific Ocean (Fig. 9). The contribution of LSF PIC (e.g., zooplanktonic foraminifera, pteropods and heteropods)
to CaCO$_3$ standing stock is higher in the subpolar gyre (35 %) than in the subtropical gyre (16 %) of the eastern North Pacific
Ocean (Ziveri et al., 2023). Betzer et al., (1984) reported that foraminifera calcite is more abundant in northern regions (north
of 42°N) of the western North Pacific. At Ocean Station Papa in the northeast Pacific (50°N, 145°W), model results showed
that foraminifera calcite accounts for only 18–30 % of the total CaCO$_3$ production, whereas coccolithophores are the major
producer, contributing up to 59–77 % of the total CaCO$_3$ production (Fabry, 1989). In the Atlantic Ocean, coccolithophore
calcite fluxes and species richness are higher in subtropical than in temperate waters, which is ascribed to the reduced
competition with diatoms in the former (Broerse et al., 2000). Based on these findings, we suggest that differences in ecosystem
structure among sites modulate the relative contribution of various calcifiers to pelagic PIC production. The higher abundance
of non-calcareous phytoplankton (e.g., diatoms) in the transition zone could also reduce coccolithophore biomass via resource
competition (Quere et al., 2005; Sinha et al., 2010) and stimulate the growth of foraminifera (Schiebel et al., 2017), resulting
in the observed decreased contribution of small coccolithophores to total CaCO$_3$ production.

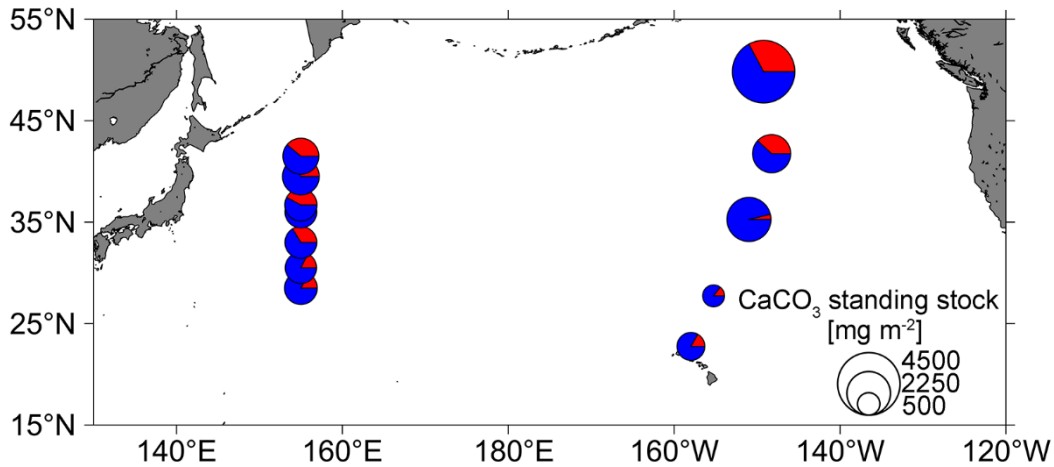

**Fig. 9.** Pie charts showing the composition of the total calcium carbonate (CaCO$_3$) standing stock in the upper 150 m of the
water column in the western (this study) and eastern North Pacific Ocean (data from the CDisK-IV cruise; Ziveri et al., 2023).
Red represents the standing stock of large size-fractionated (> 51 μm) CaCO$_3$ from this study, and planktonic foraminifera,





pteropods and heteropods from the CDisK-IV cruise. Blue represents the standing stock of small size-fractionated (1–51 μm)
CaCO₃ from this study and coccolithophores from the CDisK-IV cruise.

**4.3 CaCO₃ production compared with the eastern North Pacific**
Using a seasonal-correction method (Ziveri et al., 2023), the average coccolithophore CaCO₃ production above 150 m was
estimated to be 0.35 mol m⁻² yr⁻¹ for the entire research domain, which agrees well with the global estimate of 0.4 mol m⁻²
yr⁻¹ (Balch et al., 2007) and model result of 0.3 mol m⁻² yr⁻¹ in the North Pacific (Hopkins and Balch, 2018). In particular, this
production was 0.62 mol m⁻² yr⁻¹ in the subtropical gyre and 0.14 mol m⁻² yr⁻¹ in the Kuroshio-Oyashio transition region (Fig.
S2b). However, the latter is much lower than the recent estimate of 0.9–1.0 mol m⁻² yr⁻¹ by Ziveri et al. (2023) based on data
from the transition zone and subpolar gyre in the eastern North Pacific Ocean using the same seasonal-correction method.
Several factors may lead to the above discrepancy. First, CaCO₃ production rate on the present study was estimated based
only on coccolithophores, whereas estimates by Ziveri et al. (2023) also included the contribution from planktonic foraminifera,
pteropods and heteropods. Second, in the CDisK-IV cruise to the eastern North Pacific Ocean, coccolithophore calcite
concentrations were significantly higher than suspended seawater PIC concentrations collected by in situ pumps in the
transition zone and subpolar gyre (Fig. S5; Dong et al., 2019, 2022). Calculations based on these apparently inconsistent data
may result in an overestimation of actual CaCO₃ production. Third, high spatial and seasonal variations in PIC production
might occur between the two oceanic environments, in particular the high dynamics in the Kuroshio-Oyashio transition region
may have essentially altered the coccolithophore community and associated CaCO₃ production.
**4.4 Shallow-water CaCO₃ dissolution in the western North Pacific**
We used a box model to calculate the magnitude of metabolic CaCO₃ dissolution in shallow waters of the western North Pacific
Ocean (Eqs. 3 and 4; Dong et al., 2019, 2022). Focusing on CaCO₃ produced by coccolithophores and assuming that production
occurs in the euphotic zone, we applied the model to stations STN43 and STN45 because the calculated metabolic calcite
saturation horizon varied from 100 m to 150 m in the Kuroshio-Oyashio transition region. The results suggest that CaCO₃
might start to dissolve in setting marine particles after sinking out of the euphotic zone at the two stations, despite apparently
oversaturated ambient conditions (Fig. S6). Three different particle sinking rates (1, 10 and 100 m d⁻¹) were used in our model





calculations (Fig. 10a). To examine the possible influence by lateral transport around the Kuroshio Extension, our data in the
upper 1200 m were compared with those obtained during the CDisK-IV cruise (Fig. S5d–f). There was no significant difference
in PIC distribution patterns between western and eastern basins across 27°N–42°N in the North Pacific Ocean (one-way
ANOVA, $p > 0.05$), with the latter having relatively limited water parcel transport in the horizontal direction. We thus contend
that in our first order estimation, the effect of lateral transport on PIC distributions could be considered negligible (Dong et al.,
2019; Subhas et al., 2022).
Assuming that all coccolithophore production is exported out of the euphotic zone, our model results show that the PIC
sinking flux using the 10 m d$^{-1}$ sinking rate agrees well with data obtained from sediment traps deployed in July 1999 at a
40°N station (Honda et al., 2002) near our research domain (Fig. 10a). At this rate, the $R_{TA}$ driven by $\Omega_{met}$ showed a vertical
distribution pattern similar to TA$^*$-chlorofluorocarbon (CFC) age-based $R_{TA}$ in the entire North Pacific Ocean (Feely et al.,
2002), with both displaying a maximum at a density of 26.58 kg m$^{-3}$ corresponding to a water depth of 300 m (Fig. 10b). In
addition, our $R_{TA}$ varied within a magnitude comparable to Alk$^*$- transit time distribution (TTD) and $^{14}$C age-based $R_{TA}$ values.
Instead, those driven by ambient $\Omega$ were essentially equal to zero with densities < 27.0 kg m$^{-3}$ and became considerable with
further increasing densities below 600 m. Therefore, our model results indicate that shallow-water CaCO$_3$ dissolution indeed
occurs in the western North Pacific Ocean mainly as a result of metabolic acidification in the particulate microenvironment.



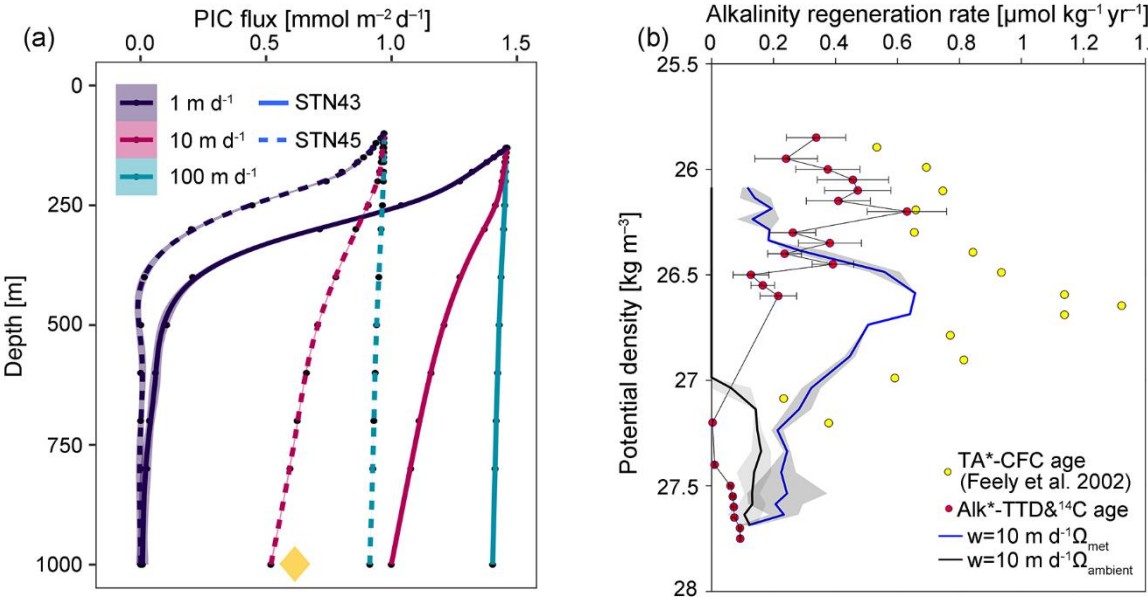


**Fig. 10.** (a) Particulate inorganic carbon (PIC) fluxes above 1,000 m at stations STN43 and STN45 generated by a box model

using three different sinking rates (1, 10 and 100 m d$^{-1}$); (b) vertical distribution of the alkalinity regeneration rate (R$_{TA}$) at

stations STN43 and STN45 generated by a 1-D model driven by metabolism-altered calcite saturation state ($\Omega_{met}$) and ambient

calcite saturation state ($\Omega_{ambient}$) at a sinking rate of 10 m d$^{-1}$. Lines indicate R$_{TA}$ binned and averaged at 0.05 kg m$^{-3}$ potential

density intervals. In (a), the yellow diamond marks the PIC flux at 1,000 m at station 40°N, 165°E in the western North Pacific

Ocean (Honda et al., 2002; Fig. 1a). In (b), TA$^*$-chlorofluorocarbon (CFC) age represents the excess alkalinity-based R$_{TA}$ from

(Feely et al., 2002) and Alk$^*$-TTD&$^{14}$C age represents the excess alkalinity-based R$_{TA}$ using transit-time distribution (TTD)

ages and $^{14}$C ages combined with Alk$^*$ (Key et al., 2004; Gebbie and Huybers, 2012; Carter et al., 2014; Jeansson et al., 2021;

Sulpis et al., 2021). Details of the estimation method are provided in the Supplementary Materials.

429

This finding is consistent with that obtained in the eastern North Pacific Ocean (Subhas et al., 2022), which suggests widely

occurring shallow-water dissolution throughout the entire North Pacific Ocean associated with organic carbon respiration. The

maximum R$_{TA}$ value of 0.66 μmol kg$^{-1}$ yr$^{-1}$ at a density of 26.58 kg m$^{-3}$ in our study aligns exactly with that of 0.6 μmol kg$^{-1}$

yr$^{-1}$ at a density of 26.54 kg m$^{-3}$ in the eastern North Pacific Ocean (Subhas et al., 2022). It is noteworthy that the R$_{TA}$



discrepancy between this and previous studies might be ascribed to various dissolution mechanisms (Fig. 10b), since our model
only accounted for coccolithophore calcite excluding other calcifying plankton groups such as those producing aragonite and
high-Mg calcite. Jansen and Wolf-Gladrow (2001) also suggested that dissolution within zooplankton guts may account for
25 % of the shallow-water dissolution signal.

**5 Conclusions**
We have demonstrated that coccolithophore abundances and compositions had distinct geographic and vertical distribution
patterns, with *U. tenuis* dominated in the NPSG region while *E. huxleyi* and *Syracosphaera* spp. in the Kuroshio-Oyashio
transition region. The environmental variables that best described varying coccolithophore communities were depth and
latitude. Calcite derived from coccolithophores contributed on average ~76 % of the PIC standing stocks above 150 m, with a
relatively greater contribution in the subtropical gyre than in the transition region. Our results suggests that coccolithophore
$CaCO_3$ production was ~5-fold higher in the former than in the latter, which highlights the importance of coccolithophores in
oligotrophic environments.
This study also inferred that extensive $CaCO_3$ dissolution occurs above the ambient calcite saturation horizon, and is
primarily driven by the metabolic activity associated with organic carbon respiration. Given the important role of $CaCO_3$
production and dissolution in the marine alkalinity and carbon cycles, additional studies are required that target
coccolithophore production at different scales from seasonal to annual and from regional to global, as well as processes leading
to $CaCO_3$ dissolution in the apparently oversaturated upper ocean.



*Data availability.* Data for temperature, salinity, coccolithophore cell and coccolith abundances, coccolithophore calcite and PIC concentrations can be downloaded from the Science Data Bank (https://www.scidb.cn/en). Satellite-based temperature, Chl *a* and PIC concentration data were obtained from the MODIS-Aqua satellite (https://oceancolor.gsfc.nasa.gov/l3/).

*Supplement link.*

*Author Contributions.* YH, ZC, and MD conceived and designed the study. YH, ZS, DF, and JC contributed to data acquisition and analysis. YH, ZS, ZC, and MD wrote the first draft of the manuscript. YH, ZS, ZC, JY, and MD discussed results and edited the paper. All authors read and approved the final version of the manuscript.

*Competing interests.* The authors declare that they have no conflict of interests.

*Disclaimer.*

*Acknowledgements.* The captain and the crew of R/V *Tan Kah Kee* are acknowledged for their cooperation during the cruise. We thank Feipeng Xu and Xin Liu for providing the chlorophyll *a* data, Lifang Wang, Tao Huang, Yanmin Wang and Zhijie Tan for the nutrient data, Yi Yang and Xianghui Guo for the carbonate system data, Xuchen Wang for advice on particulate inorganic carbon measurements, and Yanping Xu for logistical assistance. Yuye Han was supported by the Joint Training Program in Marine Environmental Sciences sponsored by the China Scholarship Council.

*Financial support.* This research was funded by the National Natural Science Foundation of China (NSFC project No. 42141003, 92258302 and 42188102). Data and samples were collected onboard the R/V *Tan Kah Kee* implementing the open research cruise NORC2022-306 supported by NSFC Shiptime Sharing Project (project No. 42149303).



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
