# Peer review of "Coccolithophore abundance and production and their impacts on particulate"

_EGUsphere, 2024_

## Author Comment (AC1)

**Response to reviewers**

We thank the reviewers for their detailed comments that helped to significantly improve the manuscript. For your convenience, reviewer comments are presented below in black font, followed by our answers in blue font.

**Reviewer #2**

The authors clearly demonstrate the substantial contribution of coccolithophores to the production of calcium carbonate standing stocks in the western North Pacific through their presentation of both Niskin bottle and size fractionated PIC measurements, as well as coccolithophore cell abundances, species compositions and diversity. The importance of metabolically driven calcite dissolution is then illustrated through the use of a simple box model, which builds upon previous studies by incorporating published data for regenerated total alkalinity and PIC flux. The results presented in this paper build upon similar studies (Ziveri et al., 2023, Subhas et al., 2022), and also integrate emerging methodologies and ideas for investigating the calcium carbonate cycle. In particular, biologically driven dissolution within the supersaturated "shallow" ocean is a quickly emerging mechanism which is helping to constrain the long-standing discrepancy between calcium carbonate production and export. Broadly, I am impressed by how the author's weaved together multiple lines of evidence to understand coccolithophores' role in the calcium carbonate cycle, especially through the lens of coccolith calcite production and dissolution. To that end, I have a few comments which I hope will help to strengthen the manuscript and improve the scope of the content.

**[Response]:** We thank Reviewer #2 for the positive and constructive feedback. We have fully considered these comments and suggestions in the revision.

The authors do a fairly thorough job of describing their assumptions, and for the most part, clarify how these translate into uncertainty within their analysis. There are a few assumptions in particular which I believe could be more thoroughly justified by the authors, 1) the assumption that satellite data underestimates PIC standing stocks, 2) the use of a range of reported PIC turnover times, and 3) the assumptions around PIC content of coccoliths, and the discussions around free liths vs coccospheres.

For the satellite imagery statement (lines 65-66) I am suggesting that you add a citation from Neukermans et al., 2023, which provides direct evidence for the discrepancy you are pointing out here. Additionally, given this stated uncertainty in satellite estimates of [PIC], I found it confusing to follow the author's justification of using satellite derived [PIC] (both from July 2022 and annual mean) to correct the PIC production rates for seasonal variability (lines 131-133). I would appreciate some commentary from the authors regarding their justification for this, for example, similar to the Supplementary Figure 2 from Ziveri et al., 2023, which illustrates the correction factor between satellite estimates and direct measurements of PIC for a given region.

**[Response]:** We appreciate your suggestions and have added the citation into the revised manuscript "This contrast, however, is smaller than the sixfold to sevenfold difference based on satellite estimates of surface PIC, indicating the importance of coccolithophore CaCO₃ production over a deeper euphotic zone and the limitations of satellite products as highlighted by Neukermans et al. (2023)".

We agree that satellite-derived PIC does not accurately represent PIC production throughout the water column. However, using satellite-derived PIC concentrations for seasonal correction is based on the

following reasons: (1) PIC production (coccolithophore production) exhibits significant seasonal variability, especially in transitional regions. Since our sampling was conducted only during the summer, it does not capture year-round production. Therefore, the seasonal corrections are essential to better represent annual patterns. (2) Although satellite-derived PIC primarily reflects PIC concentrations in the upper few meters of the water column, the relative seasonal and inter-annual variations observed at the surface broadly reflect those at depth (Neukermans et al., 2023; Ziveri et al., 2023). Therefore, they support the use of satellite-derived PIC concentrations for relative seasonal corrections, even though the absolute values may not fully represent the PIC production throughout the water column.

In response to your suggestion, we have referred to Ziveri et al. (2023), and added a supplementary figure to illustrate the correction factor between satellite-derived estimates and direct measurements of PIC for the given region.

We have clarified this issue in our revision by "Overall, our results suggest that the calibration of satellite-derived PIC is not reliable. There was a significant positive relationship between surface coccolithophores calcite concentrations and satellite-derived PIC concentrations ($r^2 = 0.84$; $p < 0.01$), which means satellite-derived PIC can reflect the distribution tendency of coccolithophore calcite concentrations but not the true values (satellite-derived PIC in high latitude area is likely overestimated, Fig S5a). Over the entire euphotic zone, our results indicate no correlation between satellite-derived PIC concentrations and actual PIC production, a finding that is also highlighted in Ziveri et al. (2023), in which the linear correlation is primarily driven by the highest data values (Fig. S5b)."

[Figure]

**Revised Fig. S5.** Scatter plots showing relationships (a) between surface coccolithophores calcite concentrations and satellite-derived particulate inorganic carbon (PIC) concentrations; (b) between coccolithophores CaCO$_3$ production rate (not seasonal corrected) in the euphotic zone and satellite-derived PIC concentrations. The red marks are data from the CDisK-IV cruise and the CaCO$_3$ production rate only includes contributions from coccolithophores (Ziveri et al., 2023).

With regards to the use of the reported range of turnover times, I think it would be worthwhile for the authors to comment on the large uncertainty that this method results in (i.e. the high production tails in Fig. 6b that stem from the order of magnitude range in turnover time estimates [0.7-10 days; Lines 124-126]), and how it may hinder interpretation of this field data. There are currently a few methods which can get at *in situ* turnover time, primarily the use of carbon isotopes spikes in incubations, with the carbon-14 method described in Graziano et al., 2000. While I completely understand that it is not possible to do this for this study, I do think it would be helpful for the authors to acknowledge that direct

measurements of turnover time could have likely reduced uncertainty within the PIC production values. Especially when considering the challenges around quantifying the oceanic calcium carbonate budget, any advancements in our ability to reduce error and uncertainty should be at the fore-front of our minds.

**[Response]:** We agree that the wide range of turnover times highly likely induces the uncertainty in our PIC production estimates, and field-based measurements of calcification rates could provide more direct and accurate value.

We have added a discussion in the revised manuscript: "While $^{14}$C incubations can provide a direct and precise measurement of in situ calcification rates, the calculation method we used offers a practical approach to convert concentration data into production estimates using turnover time (Graziano et al., 2000; Ziveri et al., 2023). This approach has limitations, particularly due to uncertainties in the estimation of coccolithophore calcite, which relies on cell counts and a morphometric-based calcite estimation method, with potential errors reaching up to 50% (Young and Ziveri, 2000b; Sheward et al., 2024). Furthermore, the calculation of production rates introduces further uncertainty, as it depends on the coccolithophore calcite standing stock and a broad range of turnover time estimates. Despite these challenges, this method produces reasonable results that are comparable to field observations and thus helps fill a critical data gap in the study region."

We have also added a sentence stating that: "In addition, more in situ calcification rate determined by $^{14}$C incubations experiments as well as direct measurements of turnover time are required to reduce uncertainty in PIC production estimations and would help in the assessment of the oceanic calcium carbonate budget."

The author's use of average coccolith/coccosphere calcite to calculate out the PIC inventory and PIC production rates could warrant more discussion on uncertainty and in general, the nuances of these assumptions. My concerns around this approach stem from the fact that Johns et al., 2023 clearly showed that the production and subsequent cycling of free coccoliths can be rapid and complex. Using a generalized [PIC] quota for each species of coccolithophore will not capture the dynamics of coccolith reabsorption, and I would suggest the authors acknowledge this. Additionally, in Lines 325-327, the authors state that "detached coccolith concentrations of… showed a significant positive relationship with their coccosphere cell concentrations, indicating that those detached particles were likely to have originated from living cells". This statement is a bit misleading, as it supposes that free liths are only coming from cells that are no longer alive. I suggest modifying in consideration of the Johns et al 2023 paper, which shows that lith shedding is a dynamic process throughout the cell's life cycle. In general, I would like the authors to comment further on the dynamics of coccolith production and shedding (i.e. how it changes with dominant species shifts, how it may be depth dependent and subsequent consequences to different dissolution processes [particle vs gut], etc).

**[Response]:** We agree that although the method for estimating coccolithophores calcite has been widely applied in many oceanic regions including the Atlantic Ocean, the Southern Ocean, and the South China Sea (Broerse et al., 2000; Jin et al., 2016; Rigual Hernández et al., 2020; Jin et al., 2022), it does have certain uncertainties, primarily from cell counts and the calcite content of specific coccolithophore species.

We have acknowledged this in our revision by stating that "We also acknowledge that estimating coccolithophore calcite and production rates using an average coccolith calcite value introduces uncertainties, as this approach does not fully account for the complexities of coccolith dynamics, including rapid cycling and reabsorption (Johns et al., 2023)."

Based on your suggestions, we have revised the relevant sentences in the revised manuscript as "It is noteworthy that detached coccolith concentrations of *E. huxleyi*, *U. tenuis* and *Syracosphaera* spp. showed a significant positive relationship with their coccosphere cell concentrations (Fig. 7b–d), indicating that these detached particles were likely shed by cells as part of the dynamic calcification process, where coccoliths are continuously produced and released (Johns et al., 2023). However, other potential sources and processes, such as advection, cell disintegration from viral lysis and grazing, fecal pellets, or the dissolution associated with microbial respiration could also contribute to the observed detached coccolith concentrations (Vincent et al., 2023; Subhas et al., 2022; Dean et al., 2024). Coccolith production and shedding are dynamic processes, with fast-growing species like *E. huxleyi* producing and shedding coccoliths rapidly during exponential growth phases, whereas other species exhibit different patterns, which are influenced by their distinct physiological and ecological characteristics (Johns et al., 2023)."

While the author's presentation of the environmental data and its influence on the coccolithophore community structure is certainly an interesting and impressive data set, I feel that it generally distracts the reader from the main goals of this manuscript. Given that the title is "Pelagic coccolithophore production and dissolution and their impacts on particulate inorganic carbon cycling in the western North Pacific", I think the authors should stick to these research aims in the manuscript, and either try to weave the environmental aspects of the study more broadly throughout the manuscript, or consider presenting the environmental-drivers as a follow-on manuscript.

[Response]: Thank you for your insightful feedback on the manuscript's aims and structure. Considering the recommendations from Reviewer #1, we have reorganized the discussion section to streamline the scope of the paper focusing on coccolithophore calcite production. As a result, we have removed Section 4.4 on shallow-water dissolution of $CaCO_3$. The revised discussion section is as follows:

4.1 Contribution of coccolithophore calcite to PIC

4.2 Coccolithophore responses to environmental factors (Per your suggestion, we have weaved the environmental aspects more broadly. Please also see our response to relevant comments by Reviewer #1)

4.3 $CaCO_3$ production compared with the eastern North Pacific

I appreciate the author's approach to assessing the role of metabolically driven dissolution, as this is an emerging mechanism impacting the shallow calcium carbonate cycle that has major implications for the field. I have some suggestions for interpreting the box-model dissolution output against previously reported observations of alkalinity regeneration (presented in Fig 10b). I was surprised the authors did not comment on the mismatch between the maxima of alkalinity regeneration between observations and model output. The observations show a clear peak alkalinity regeneration well above 500m, whereas the model output shows the peak occurring at or near 500m. I believe this mismatch is due to unique "metabolic" processes that occur at different depths in the water column. For example, micro- and macro-zooplankton primarily graze within that upper 500m, and therefore, could be the primary drivers of that shallow alkalinity regeneration due to calcium carbonate dissolution during ingestion and digestion processes. A recently published paper (Dean et al., 2024) titled provides experimental evidence that microzooplankton facilitate a substantial amount of coccolith calcite dissolution through grazing and digestion. I bring this up, because the model, and broader context of this discussion, is largely focused on particle associated metabolic dissolution. If the authors were to consider the different types of metabolic dissolution that occur at different depths in the water column, they may be able to directly

comment on the maxima mismatches in Fig 10b and provide a stronger discussion for section 4.4. This is an active area of research which warrants a deeper discussion from the authors.

[Response]: Thank you for pointing out the important issue of the mismatch between observations and model outputs in Fig. 10b. The observations represent alkalinity regeneration over a broad latitudinal range of 30 °N–45 °N, while the model results are specifically derived from two stations around 40 °N. This geographic difference may partially account for the mismatch. Additionally, while the two isopycnal surfaces ($\sigma_\theta$=26.1 and 26.5) correspond to different densities, they are both situated at approximately 300 m, indicating that the maxima of alkalinity regeneration are geographically and vertically close. However, we agree that density-specific metabolic processes, particularly macro-zooplankton grazing and digestion, play an important role in driving shallow $CaCO_3$ dissolution, whereas zooplankton grazing and free coccolith contributions were not explicitly included in our model.

Please note that mainly based on Reviewer #1's suggestions, we have removed this entire dissolution section in the revised manuscript. However, your insightful comments on this part will be incorporated into our future research.

Overall, there is a lot of valuable and needed field data being presented in this manuscript which certainly contributes to our understanding of calcium carbonate cycling dynamics, especially as driven through coccolithophores. My overall suggestion to the authors is to really hone in the scope of this manuscript, and keep the focus to coccolith calcite production and dissolution. The environmental data, while incredibly interesting, could enhance the scope of this study if integrated into the discussion further, or would serve well being presented independently as a follow-on study.

[Response]: We greatly appreciate your recognition of the value of the field data presented in our manuscript and its contribution to understanding $CaCO_3$ cycling driven by coccolithophores.

Comprehensively considering comments from both reviewers, we have reorganized the manuscript to focus on coccolithophore calcite production, mainly by removing the section of shallow-water dissolution of $CaCO_3$ and strengthening the discussion regarding environmental data.

Additional minor comments:

● I might suggest modifying the title to be "Coccolithophore production and dissolution and their impacts on pelagic particulate inorganic carbon cycling in the western North Pacific", given that "pelagic" is really in reference to the depths of PIC cycling, and not coccolithophore PIC production, which is always pelagic.

[Response]: "Pelagic" has been removed. The new title is "Coccolithophore abundance and production and their impacts on particulate inorganic carbon cycling in the western North Pacific".

● Line 59, could expand from just "zooplankton" to "micro and meso zooplankton". Could also add a citation for my recent publication (Dean et al., 2024) which provides direct experimental evidence for microzooplankton facilitated dissolution, and shows that microzooplankton food vacuoles are acidic and eventually buffered from coccolith calcite dissolution.

[Response]: We have removed this paragraph due to the removal of Section 4.4, as Reviewer #1 suggestion.

● Citation for lines 65-66, regarding satellite PIC and measured/integrated PIC discrepancies? Suggestion is Neukermans et al., 2023 (see end of review for reference list)

**[Response]:** The citation of Neukermans et al. (2023) has been added.

- Line 119-120: Considerations around using an average coccolith calcite mass and potential error?

  While most other stated assumptions are followed with a comment on the uncertainty they bring, I found this assumption to be not acknowledged at the same level as others. Would suggest adding a statement which at least speculates on the uncertainty this is adding to your calculations. For example, different coccolithophore species have very differing growth rates, and consequently, different PIC production rates/turnover times. Given the marked differences in coccolithophore species composition reported in this study, I believe the authors should expand upon this section.

**[Response]:** Based on your suggestion, we have added the following statement in the revised manuscript:

**1. Methods section:** "The coccolithophore turnover time was derived from both laboratory and field estimates, as well as simulations from a generalized coccolithophore model, which has also been applied to the eastern North Pacific (Krumhardt et al., 2017; Ziveri et al., 2023). We are aware that different coccolithophore species exhibit widely varying growth rates, and cell growth phase differs for smaller cells produce fewer coccoliths during the exponential growth phase (characterized by rapid division), whereas larger cells generate more coccoliths during the early stationary phase, when cell division slows down (Raven and Crawfurd, 2012; Krumhardt et al., 2017). We also acknowledge that estimating coccolithophore calcite and production rates using an average coccolith calcite value introduces uncertainties, as this approach does not fully account for the complexities of coccolith dynamics, including rapid cycling and reabsorption (Johns et al., 2023). Despite these possible errors and uncertainties, our estimations are consistent with direct production rate measurements of Daniels et al. (2018), suggesting that our data provide a reliable basis for assessing coccolithophore calcification dynamics."

**2. Discussion section:** "While $^{14}$C incubations can provide a direct and precise measurement of in situ calcification rates, the calculation method we used offers a practical approach to convert concentration data into production estimates using turnover time (Graziano et al., 2000; Ziveri et al., 2023). This approach has limitations, particularly due to uncertainties in the estimation of coccolithophore calcite, which relies on cell counts and a morphometric-based calcite estimation method, with potential errors reaching up to 50% (Young and Ziveri, 2000a; Sheward et al., 2024). Furthermore, the calculation of production rates introduces further uncertainty, as it depends on the coccolithophore calcite standing stock and a broad range of turnover time estimates. Despite these challenges, this method this method produces reasonable results that are comparable to field observations and thus helps fill a critical data gap in the study region."

- Line 230-231: Clarification Question, is this in reference to all *E. huxleyi* associated calcite (i.e. coccospheres and liths)? Or just coccospheres?

**[Response]:** Only *E. huxleyi* coccospheres here. We have clarified this point by adding "The coccospheres of *E. huxleyi*".

- Fig 5. Suggestion for readability: Add labels to the pie charts to make it easiest to compare coccosphere to liths to total calcite.

**[Response]:** Figure 5 has been redrawn in the revised manuscript.

[Figure]

**Revised Fig. 5.** Contribution of different coccolithophore groups to coccosphere cell abundance, detached coccolith abundance, and coccolithophore calcite concentration in the upper 300 m of the water column: (a–c) in the North Pacific Subtropical Gyre (NPSG: M30, M32 and M35) and (d–f) in the Kuroshio-Oyashio transition region (KE3, STN41, STN43 and STN45). Lower euphotic zone (LPZ) species include *Florisphaera profunda* and *Algirosphaera robusta*.

● Line 243: Why a depth of 150m for standing stock integration? I don't think this was stated in the methods or results. Can you comment on the implications of setting the same depth for this integration? **[Response]:** Based on your suggestion, as well as comments from Reviewer #1, we have revised the integration depth from 150 m to the euphotic zone depth as determined by the 0.1% surface PAR. However, this adjustment does not affect the main conclusions.
Please also see our response to relevant comments by Reviewer #1.

● Fig 6b. I had never seen a violin plot before, so it was a bit difficult for me to interpret this figure on my first pass. I would suggest that the authors provide additional description in the caption to at least describe what the blue diamond, blue line, and shape of the plots represent.
**[Response]:** We have replotted Figure 6 and rewritten the caption for clarification.

[Figure]

**Revised Fig. 6.** Calcium carbonate ($CaCO_3$) standing stock in the euphotic zone estimated from Niskin bottle particulate inorganic carbon (PIC), total calcite (Cocco) and size-fractionated (large and small fractions indicate > 51 and 1–51 μm, respectively) PIC concentrations (a) at each sampling station and (b) in the North Pacific Subtropical Gyre (NPSG) and Kuroshio-Oyashio transition regions; (c) $CaCO_3$ production by coccolithophores in the euphotic zone at indicated sampling stations in June-July 2022; (d) annual $CaCO_3$ production corrected for seasonal bias using satellite-derived PIC concentrations. The blue diamond marks the median value, while the shaded area displays the probability density of the estimates. The grey lines denote the quartiles (the 25th and 75th percentiles).

● Line 292: RDA has not been previously defined. Only defined in Fig. 7 caption, so you may want to define it in the text. Additionally, I think the authors could provide additional justification for their choice in using RDA for this study. I would suggest adding this in the methods section, so that someone who is less familiar with multivariate analyses can understand why this is appropriate here.

**[Response]:** We have added the corresponding content in the methods section as "The redundancy analysis (RDA) is a widely used multivariate analytical method to identify relationships among individual variables in different categories. Prior to the RDA, statistical differences in environmental

variables were evaluated using an analysis of variance (one-way ANOVA), while collinearity between environmental variables was accounted for by calculating variance inflation factors (VIF). Forward selection of variables was subsequently carried out until all VIF scores were <10, in order to only include variables that are not significantly correlated. These criteria reduced the number of environmental variables used in the RDA. Monte Carlo permutation tests, based on 1000 randomizations, were performed to identify the most significant and independent effects on variation in the composition of the coccolithophore community. The overall significance of the explanatory variables after forward selection was evaluated through ANOVA ($\alpha$<0.05) and coefficient of determination ($r^2$) and adjusted $r^2$ were calculated to assess the power of a selected RDA model using the vegan package (Oksanen, 2010)."

● Fig 8a. Clarifying, does coccolithophore calcite = coccosphere and coccoliths? Or just coccosphere?

**[Response]:** Here, coccolithophore calcite refers to the total calcite content, which includes both the calcite within coccospheres and the detached coccoliths. We have clarified this point in our revision.

● Line 333: what is the "aggregation group"? Can you please explain further?

**[Response]:** In the scanning electron microscope (SEM) analyses, "aggregation groups" refer to clusters formed by multiple coccolithophores grouped together. In the study, we only quantified their abundance but excluded them from coccolithophore calcite calculations, as it is challenging to accurately determine the number of individual coccoliths within these aggregates.

We have added details in the "methods" section as "Aggregates formed by clusters of multiple coccolithophores were quantified in terms of abundance but were excluded from the coccolithophore calcite calculations, mainly due to the difficulty in accurately determining the number of individual coccoliths within the aggregates."

● Lines 339-341: "Thus, although E. huxleyi is one of the most abundant species in the ocean, larger coccolithophore species can also play an important role in CaCO$_3$ export."

I would like the authors to provide some justification for this statement from the data presented in this manuscript. It seems that Fig 5. Could be modified to illustrate this point, perhaps by adding percentages and errors to the pie charts?

I don't think Fig 5 does a good job of illustrating this point, since they are grouped into the "other" category. I would suggest having a supplementary figure which shows the contributions of these larger species to the total inventory, so that you might reference them in this statement.

**[Response]:** We have revised this figure to better illustrate the differences in the contributions of various coccolithophore groups to coccosphere cell abundance, detached coccolith abundance, and coccolithophore calcite concentration between the NPSG region and the Kuroshio-Oyashio transition region. Percentage values have been added. We have also explicitly included *Calcidiscus leptoporus* and *Oolithotus fragilis* in the revised figure. Although these species are less abundant, they contributed significantly to the coccolithophore calcite. We believe these revisions more effectively highlight the role of larger species in CaCO$_3$ production and export.

Additionally, we have revised the text as: "The less abundant (<3 %) species such as *C. leptoporus* and *O. fragilis* also made a large contribution to calcite concentrations, accounting for 21 % and 12 % of the coccolithophore calcite concentration in the NPSG region and the Kuroshio-Oyashio transition region, respectively (Fig. 5)."

Please refer to our earlier response regarding the revised Figure 5.

- Line 345-347: Does this need a citation?

  With respect to the DCM being deeper than 100m in subtropical gyres

  **[Response]:** The citation of Cornec et al. (2021) has been added.

- Lines 347-348: Can the authors provide a value or range of values for the underestimation here?

  **[Response]:** Sorry, we are unable to provide a specific value or range of values for the underestimation. Instead, we have revised the text to make this point more accurate: "In these oligotrophic and low productivity oceans, a subsurface PIC maximum can develop within the euphotic zone, and the highly variable subsurface PIC concentrations are poorly reflected by satellites, potentially limiting the ability to fully capture coccolithophore contributions."

- Lines 369-372: Did the authors look at any metrics for the composition of the non-coccolithophore community members? Since there is a point here about competition, and the authors did such a thorough job of investigating the coccolithophore diversity, it would be interesting to extend that thought to the results of this study.

  **[Response]:** While we do not have data on non-calcareous phytoplankton abundances, we have added references of previous studies observing latitudinal gradients in diatom biomass and planktonic foraminifera abundance (Hirata et al., 2011; Sugie and Suzuki, 2017; Taylor et al., 2018). Furthermore, we have included supporting evidence from sediment trap data in the North Pacific, which indicate lower fluxes of planktonic foraminifera, organic matter, and biogenic opal in the subtropical region than in the transitional and subarctic regions (Eguchi et al., 2003). These additions help to better discuss the influence of ecosystem structure on the relative importance of different calcifiers across regions.

- Line 394: "high dynamics" seems vague to me, but perhaps I'm misinterpreting this?

  **[Response]:** We have replaced "high dynamics" with "the complex environmental gradients and variability".

- Line 401: I believe there is a typo in this sentence? "The results suggest that CaCO3 might start to dissolve in *"setting"* marine particles after sinking out of the…" Should *"setting"* be *"settling"*?

  **[Response]:** Yes, it is a typo. This sentence has been removed.

- Line 436-437: Could cite my recent publication (Dean et al 2024) to show that MZP also contributes substantially to dissolution in lab study.

  **[Response]:** The reference of Dean et al. (2024) has been cited elsewhere necessary in the revised manuscript.

[revised manuscript text omitted]

---

## Author Response (AR2)

*Re: "Coccolithophore abundance and production and their impacts on particulate inorganic carbon cycling in the western North Pacific" by Yuye Han et al.*

May 2, 2025

Dear Editor,

Thank you very much for your decision to accept our manuscript, "Coccolithophore abundance and production and their impacts on particulate inorganic carbon cycling in the western North Pacific", for publication in *Biogeosciences*. We are sincerely grateful to you and the reviewers for your time and thoughtful feedback on our revised manuscript.

In response to the most recent comments, we have made further revisions to our manuscript. Below, we provide a detailed summary of the changes made.

Sincerely,

Minhan Dai & Zhimian Cao

State Key Laboratory of Marine Environmental Science

Xiamen University, Xiang'an District, Xiamen 361102, China

E-mail: mdai@xmu.edu.cn & zmcao@xmu.edu.cn

**Response to reviewers**

Ln 22, it would be good to define small PIC (i.e. <51 um) here in the abstract as it is not explained until later in the manuscript.

[Response]: We have now defined "small-size fraction (1–51 μm)" and "total PIC (> 1 μm)" in the abstract of the revised manuscript (see Lines 21–22).

Ln 234, species name should be *Discosphaera tubifera* rather than *Dicosphaera*.

[Response]: We have corrected this in the revised manuscript (see Line 234).

Ln 469, Consider rephrasing the line 'our results suggest that calibration of satellite-derived PIC should be unreliable'. Do the authors mean calibration (i.e. value to value comparison) or interpretation of patterns and dynamics of satellite-derived PIC? The thrust of the discussion is that more is happening at depth in terms of PIC production and species dynamics than is revealed in satellite-derived PIC (as it sees only the surface waters) rather than satellite PIC being a poor measure of in situ PIC.

[Response]: Following the suggestions from the Reviewer, we have rephrased the whole paragraph as follows:

Overall, our findings suggest that while satellite-derived PIC can reflect surface-layer distribution patterns, its calibration should be interpreted with caution, as it does not reliably capture total water column PIC production. We observed a significant positive relationship between surface coccolithophore calcite concentrations and satellite-derived PIC concentrations ($r^2 = 0.84$; $p < 0.01$; Fig. S5a), indicating that satellite data can reflect the spatial distribution trends of the surface calcite. However, this correlation does not extend to actual values, particularly in high latitude areas where satellite-derived PIC is likely overestimated. Across the full euphotic zone, no significant correlation was found between satellite-derived PIC and measured PIC production, which is also noted by Ziveri et al. (2023) for the CDisK-IV cruise (Fig. S5b). More in situ measurements, such as calcification rates determined from $^{14}$C incubations and direct measurements of coccolithophore turnover time, are needed to reduce uncertainties in estimating PIC production and assessing the oceanic $CaCO_3$ budget. Please see Lines 469–477 of the revised manuscript.

**Reference**

Ziveri, P., Gray, W. R., Anglada-Ortiz, G., Manno, C., Grelaud, M., Incarbona, A., Rae, J. W. B., Subhas, A. V., Pallacks, S., and White, A.: Pelagic calcium carbonate production and shallow dissolution in the North Pacific Ocean, Nat Commun, 14, 805, https://doi.org/10.1038/s41467-023-36177-w, 2023.